# Investigating Brain Microstructural Alterations in Type 1 and Type 2 Diabetes Using Diffusion Tensor Imaging: A Systematic Review

**DOI:** 10.3390/brainsci11020140

**Published:** 2021-01-22

**Authors:** Abdulmajeed Alotaibi, Christopher Tench, Rebecca Stevenson, Ghadah Felmban, Amjad Altokhis, Ali Aldhebaib, Rob A. Dineen, Cris S. Constantinescu

**Affiliations:** 1Division of Clinical Neuroscience, Nottingham University Hospitals NHS Trust, School of Medicine, University of Nottingham, Nottingham NG7 2UH, UK; Christopher.Tench@nottingham.ac.uk (C.T.); Rebecca.Stevenson@nottingham.ac.uk (R.S.); Ghadah.Felmban@nottingham.ac.uk (G.F.); Amjad.Altokhis@nottingham.ac.uk (A.A.); rob.dineen@nottingham.ac.uk (R.A.D.); Cris.Constantinescu@nottingham.ac.uk (C.S.C.); 2School of Applied Medical Sciences, King Saud bin Abdul-Aziz University for Health Sciences, Riyadh 14611, Saudi Arabia; aldhebaiba@gmail.com; 3School of Health and Rehabilitation Sciences, Princess Nourah bint Abdulrahman University, Riyadh 11564, Saudi Arabia; 4NIHR Nottingham Biomedical Research Centre, Nottingham NG1 5DU, UK

**Keywords:** diabetes mellitus, type 1 diabetes, type 2 diabetes, diffusion imaging, Diffusion Tensor Imaging (DTI), microstructural abnormalities, cognitive dysfunction

## Abstract

Type 1 and type 2 diabetes mellitus have an impact on the microstructural environment and cognitive functions of the brain due to its microvascular/macrovascular complications. Conventional Magnetic Resonance Imaging (MRI) techniques can allow detection of brain volume reduction in people with diabetes. However, conventional MRI is insufficiently sensitive to quantify microstructural changes. Diffusion Tensor Imaging (DTI) has been used as a sensitive MRI-based technique for quantifying and assessing brain microstructural abnormalities in patients with diabetes. This systematic review aims to summarise the original research literature using DTI to quantify microstructural alterations in diabetes and the relation of such changes to cognitive status and metabolic profile. A total of thirty-eight published studies that demonstrate the impact of diabetes mellitus on brain microstructure using DTI are included, and these demonstrate that both type 1 diabetes mellitus and type 2 diabetes mellitus may affect cognitive abilities due to the alterations in brain microstructures.

## 1. Introduction

Diabetes Mellitus (DM) is a metabolic disorder characterised by persistent high blood glucose levels (hyperglycaemia), and inadequate insulin production or insulin dysfunctionality. This disorder is classified into two types with different mechanisms known as type 1 diabetes mellitus (T1DM) and type 2 diabetes mellitus (T2DM) [1]. Both types of DM may have an impact on brain functions and microstructure, and this has become a topic of interest for many researchers in the neuroimaging field. T1DM is associated with a modest decline in cognitive ability and reductions of brain volume in several regions, which are most noticeable in T1DM patients with an early childhood onset of diabetes [2,3]. However, in adults, the reductions in volume and the microstructural abnormalities may occur after prolonged exposure to diabetes [4]. A previous meta-analysis has shown worse performance in multiple cognitive domains in T1DM due to the brain issue abnormal changes, including intelligence, information processing speed, psychomotor efficiency, visual and sustained attention, visual perception, cognitive flexibility and executive functions [2,3,4,5].

On the other hand, T2DM has an association with general brain atrophy and microstructural changes that may be caused by microvascular complications and ageing [6]. The microstructural abnormalities are seen in the white matter; when occurring in the temporal lobe, these were associated with poor memory performance [7,8], whereas abnormalities in frontal and temporal lobes were associated with a reduced speed of information processing [7]. Additionally, T2DM is found to be associated with a modest decline in the domain of executive functioning [9].

Diffusion Tensor Imaging or DTI is a quantitative method that can be used to reveal microstructural alterations in cerebral structures; mainly, in the tracts of the white matter. DTI detects these changes before anatomical abnormalities may occur and become obvious on conventional structural MRI scans [10,11,12]. DTI quantitively measures the motion of water molecules within the brain tissues and can be sensitive to axonal and water orientations in intracellular and extracellular compartments [13]. The water molecules in free spaces diffuse uniformly in many different orientations, which is known as isotropic water motion. In contrast, the movement of water molecules becomes oriented to a specific direction when cellular barriers are present, such as cellular membranes and myelin sheaths, known as anisotropic diffusion [14,15]. DTI has primary and supplementary metrics for evaluating the diffusivity of water molecules in the white matter microstructure. Fractional anisotropy (FA) is a primary parameter which increases with the coherence of fibre tracts and myelination and decreases with extracellular water diffusion [16,17]. Axial diffusivity (AD) and radial diffusivity (RD) are additional metrics that quantify the water diffusion of molecules perpendicular and parallel to the dominant nerve fibre orientation, respectively [11]. Mean diffusivity (MD) is also a primary metric that calculates the overall diffusivity of water molecules and reports the average effects of diffusivity changes in white matter [16,17]. These DTI metrics have different indications, lower FA, and higher MD values, and they are mostly indicative of a reduction of microstructural tissue integrity. In contrast, altered AD values reflect axonal damage, and altered RD primarily indicates the integrity of the myelin sheath [18]. However, in some cases, the DTI metrics may reflect inaccurate levels of water molecules diffusivity, and that can be due to neuronal inflammation or the existence of crossing fibres [17,19].

Herein, we aim to investigate the neurological impact of T1DM and T2DM on brain microstructure and the correlation with cognitive functions using DTI. In this review, the brain DTI studies conducted on patients with T1DM and T2DM have been systematically reviewed; the following are discussed: microstructural abnormalities and the relationship with cognitive performance and metabolic profile.

## 2. Materials and Methods

This review was registered in the International Prospective Register of Systematic Reviews (PROSPERO) database (CRD42020177420) and conducted according to the Preferred Reporting Items for Systematic reviews and Meta-Analysis (PRISMA) guidelines.

### 2.1. Data Sources and Search Strategy

Three electronic databases (Medline, Embase and PubMed) were searched from April 2020 to October 2020 for identifying the relevant studies. A librarian developed the search terms and strategies, and the databases were identified and selected by A.A. and R.S. Mendeley library was used to organise and manage the identified studies. The screening (titles, abstracts and full texts) was conducted by four specialised reviewers (A.A, G.F, C.T and A.T) independently and blindly. For eligible studies inclusion, all references listed in the included studies were reviewed carefully by A.A, R.D, and C.S.C, A.D reviewed the entire content. Finally, disagreements between all reviewers were solved by a mutual discussion.

### 2.2. Study Selection and Inclusion Criteria

Studies in this review were selected and included based on an agreed list of inclusion and exclusion criteria (Table 1). All articles written in a language other than English or involving animals were excluded from this review. The age range of participants in all included studies had to be between 5 and 75 years. Studies which recruited patients with T1DM and T2DM were included only. However, studies which recruited patients who had neurological/metabolic disorders other than diabetes, cancer, brain implants and surgeries were excluded.

### 2.3. Data Extraction and Management

In the middle of July 2020, data from all included studies were extracted and organised using Microsoft Excel. Three supplementary tables of each diabetes group were designed to extract the data ideally. In the first table, the demographic and clinical characteristics of patients and healthy individuals were obtained separately. In the second table, the DTI analysis and between-group diffusion findings; specifically, the microstructural changes and DTI metrics were extracted. Lastly, the associations of diffusion metrics with cognitive dysfunctions and metabolic profile were extracted in the third table.

## 3. Results

The electronic search of the three databases retrieved 649 studies, and after removing the duplicated studies, a total of 519 studies remained. The title and abstract screening excluded 394 studies that did not meet the eligibility criteria. After the title and abstract review, 125 studies remained for full-text assessment. Following the full-text evaluation, 87 studies were excluded based on the reasons shown on the PRISMA diagram. A total of thirty-eight studies that had explored the brain microstructure in type 1 and type 2 diabetic patients using DTI were included as they met the inclusion criteria of this review. The detailed process of article searching, identification and screening can be found in Figure 1.

### 3.1. Overview of Reviewed DTI-T1DM Studies

A total of nine DTI-T1DM studies investigating the brain microstructural changes and their relationships with cognitive functions and metabolic profile in T1DM were included in this comprehensive review. The studies were designed cross-sectionally, and most of them were performed in the United States between 2008 and 2018. The participants were young adults and individuals in their teens, except for two studies [20,21] which were conducted on children under ten years of age (Figure 2a). With regard to the clinical characteristics, the mean disease duration, the mean of haemoglobin A1c (Hb1Ac), and the mean of fasting blood sugar (FBS) in the included studies are summarised in (Figure 2b–d).

A total of five included studies examined the DTI diffusion parameters such as MD, FA, AD or RD, while the other four articles only investigated the white matter using FA [2,22,23,24]. Three different methods of analysis were used; however, tract-based spatial statistics (TBSS) analysis was used frequently for the whole brain (Figure 3). A total of six studies acquired the data on 1.5T MRI scanners, and three studies obtained the images on 3T MRI units [2,21,24].

#### 3.1.1. Between-Group Differences: Microstructural Alterations in T1DM

A total of eight of the included studies consistently revealed the reduction of FA in T1DM [2,4,21,22,23,24,25,26], while four studies revealed higher MD, RD and AD in the white matter of T1DM patients compared to healthy controls (Table 2) (Figure 4) [4,20,21,26]. Most of the studies analysed DTI images using only voxel-wise brain analysis while other studies used both region of interest (ROI) and voxel-wise analysis [22,26], except for one study which used DTI tractography [24].

Using the TBSS method for analysing the whole-brain white matter, studies reported lower FA in T1DM patients compared to healthy participants in different structures. van Duinkerken et al. reported a widespread reduction of FA in T1DM patients with and without microangiopathy compared to controls [8]. In T1DM patients with microangiopathy, reduced FA was observed bilaterally in the thalamic radiation, corpus callosum, superior longitudinal fasciculus and forceps minor, while reduced FA in patients without microangiopathy was limited to the corpus callosum and right corona radiata [4]. Ahmed et al. reported a reduction of FA in the white matter tracts of adolescents with T1DM; this decreased FA was mainly observed in optic radiation and superior longitudinal fasciculus [9]. However, Kodl et al. reported reduced FA in the optic radiation, posterior corona radiata, splenium and the posterior body of the corpus callosum, which partially overlaps with the study by van Duinkerken et al., who reported lower FA in the corpus callosum and right corona radiata of T1DM patients without microangiopathy [7,8]. A widespread reduction in FA was observed in T1DM patients with younger age onset [21]. In middle-aged patients, Franc et al. reported that the subjects with T1DM demonstrated significant lower measures of FA than the control subjects; specifically, in the optic radiation [3]. Yoon et al. reported an extended observation of FA reduction in the white matter of young adult patients, where reduced FA was found bilaterally in the superior longitudinal fasciculi, corpus callosum, and right inferior frontal-occipital fasciculus [5]. Antenor-Dorsey et al. found variations in MD and RD values in adolescents with T1DM. However, lower AD was significantly reported in the superior parietal lobule, corpus callosum, external capsule, posterior limb of the internal capsule and cerebral peduncle compared with healthy volunteers [4]. Barnea-Goraly et al. examined the effects of disease onset and duration in children with T1DM. The results showed no significant differences in MD but a widespread reduction in RD in the white matter and reduced AD throughout patients’ brains; significantly, in patients with longer disease duration. Contrarily, the study reported a general increase in RD in the white matter of patients with younger age at disease onset [6]. In middle-aged adults with T1DM, there were no significant MD differences reported. However, a substantial increase in RD was in the posterior portion of the corpus callosum, bilateral corticospinal and inferior frontal occipital tracts, forceps minor and corona radiata in T1DM patients with and without microangiopathy. In addition, the authors observed significantly lower AD in the corpus callosum, forceps minor and major, superior longitudinal fasciculus and thalamic radiation in patients with and without microangiopathy [4]. Aye et al. observed no significant RD differences in children with T1DM, which may be due to short disease duration; however, a considerable decrease in AD was found in the internal capsule, the body of the corpus callosum, right cingulate gyrus, right thalamus, right superior temporal gyrus, and posterior parietal lobe [1].

Using DTI tractography method, Franc et al. investigated grey matter in middle-aged adults with T1DM and reported a significant reduction of cortical thickness for cortical regions with high connectivity to the optic radiations and posterior corona radiata tracts [3]. Using the ROI-analysis method, two studies reported a pronounced FA reduction in the white and grey matters of adolescents with T1DM compared to controls. Antenor-Dorsey et al. noticed lower FA in superior parietal lobule; likewise, Toprak et al. observed FA reduction in various regions of T1DM brain such as in the putamen, inferior longitudinal fasciculus, thalamus, corticospinal tracts at the pons, frontal white matter, hippocampus and corona radiate [2,4]. In adolescents with T1DM, only one study investigated the between-group changes of the additional diffusion indices such as MD, AD and RD using ROI analysis method. The study significantly reported a reduction of MD in the thalamus and an increase in RD in the cerebellum; however, lower AD was observed in the cerebellum, and no FA, MD, RD and AD differences were seen between groups in the hippocampus [26].

#### 3.1.2. Microstructural and Cognitive Abnormalities in T1DM

The findings of the reviewed studies suggest that brain changes caused by T1DM have an association with attention deficits, information processing, psychomotor speed, and executive functions. In adolescents with T1DM, Ahmed et al. reported that the score test of Wisconsin Card Sorting Test was positively correlated with FA at the midbrain region, and this correlation showed worse performances on tests for attention, speed of information processing, and executive function [9]. The Benton Visual Retention Test was used to test visual perception, visual memory, and constructive visual abilities, which showed a positive relationship between impairment and changes in corona radiata, cingulum, splenium and optic radiation [23]. In contrast, a negative correlation between the intelligence quotient (IQ) level and FA of mid-brain, thalamus and the posterior portion of the internal capsule with verbal point [22].

In children with T1DM, a correlation between lower FA and Full-Scale Intelligence Quotient (FSIQ) approached significance in the bilateral parietal lobes and the right superior temporal gyrus, which caused modest alterations in cognitive abilities such as visuospatial abilities, attention, general intellectual ability, memory, processing speed, and executive function [21]. Another study reported a positive correlation between the Wechsler Intelligence Scale for Children and FSIQ with FA values and information processing speed [20].

In young adults with T1DM, Franc et al. found that there was a significant relationship between the performance and cortical thickness in posterior grey matter regions, which suggested the involvement of these regions in executive function and psychomotor speed [3]. Recruiting the same subjects, Kodl et al. confirmed that reduced FA values in the white matter correlated with poor cognitive performance [7]. Lastly, Yoon et al. noted that lower working memory performance associated with low FA values in the bilateral superior longitudinal fasciculi in young adults with T1DM (Figure 5) [5].

#### 3.1.3. Microstructural Alterations and Metabolic Profile in T1DM

The findings of included studies suggest that longer disease duration and increased HbA1c values correlated with the severity of microstructural abnormalities in optic radiation [2,23], posterior corona radiata [2], right internal capsule, anterior forceps, inferior frontal occipital fasciculi, splenium of the corpus callosum [2], uncinate fasciculi, subgenual white matter, superior, middle and inferior temporal gyri, superior longitudinal fasciculi, and occipital white matter [20,21,25].

In adolescents, Ahmed et al. reported an inverse relationship between duration of diseases and lower FA in optic radiation only [9]. However, Barnea-Goraly et al. found that the increased duration of T1DM was positively associated with FA in the occipital, frontal, temporal, parietal lobes and multiple white matter tracts, this is likely because of the younger age and the effect of age in this study [6]. Additionally, the increased duration of T1DM was associated with reduced AD and RD in the younger age of T1DM onset. HbA1c values at baseline were negatively associated with FA values and positively associated with RD values [21].

In middle-aged adults, reduced FA values in the posterior corona radiata, the optic radiation, splenium and the posterior body of the corpus callosum were correlated with the duration of diabetes and the increased values of HbA1c [2]. In young adults, there was a negative relationship between FA values and current HbA1c. However, long-lasting glycaemic control levels, severe hypoglycaemia, and the duration of disease were not correlated with FAs in the T1DM group [25]. 

In children, Aye et al. described a negative correlation between HbA1c values and FA in the anterior forceps, right internal capsule, inferior frontal occipital fasciculi, and splenium of the corpus callosum [1]. Conversely, a positive correlation was reported between HbA1c and RD in inferior frontal occipital fasciculi, uncinate fasciculi, subgenual white matter, anterior forceps, the right internal capsule, superior middle and inferior temporal gyri, the splenium of the corpus callosum, superior longitudinal fasciculi, and occipital white matter [20] (Figure 6a–c).

### 3.2. Overview of Reviewed DTI-T2DM Studies

A total of twenty-nine DTI-T2DM studies investigating the brain microstructural changes and their relationships with cognitive functions and metabolic profile in T2DM patients were included in this review. All of these studies were designed cross-sectionally, and most of the included studies were performed in China and the United States since 2016. The participants were mostly adults in their middle age or older, except for three DTI-T2DM studies which were conducted on young adult patients [8,28,29] (Figure 7a). With regard to the clinical characteristics, the mean disease duration, the mean of haemoglobin A1c (Hb1Ac), and the mean of fasting blood sugar (FBS) in the included studies are summarised (Figure 7b–d).

A total of twenty-five of the reviewed articles examined DTI diffusion indices such as FA, AD, RD, or MD, while four other studies utilised graphical theory to analyse the network of white matter using the DTI technique. The TBSS technique of the whole brain was used more frequently in the reviewed studies, and FA maps were often investigated (Figure 8). In the included studies, most of the data were acquired on 3T MRI scanners, while few other datasets were obtained on 1.5T MRI units.

#### 3.2.1. Between-Group Differences: Microstructural Alterations in T2DM

Most of the reviewed studies consistently revealed lower FA and higher AD, MD and RD in the white and grey matter using voxel-wise whole-brain or global brain analyses (Table 3) (Figure 9). Methods of analysis used in these studies included TBSS, tractography and voxel-based methods of analysis.

Most of the findings were reported in the frontal and temporal lobes in T2DM patients [8,29,30] and also reported the differences between groups of additional diffusion parameters such as RD and AD. Some other studies reported increased diffusion of the additional parameters such as AD and RD in the regions with low FA values or high MD values. However, different methods of the analysis showed changes only in RD values [29,31,32,33], or in AD only [34,35]. Disruption in the brain network using graph theoretical network analysis was described by network integrity parameters, which is characterised by the lower overall efficiency and the length of the path in all four studies [31,36,37,38]. The parameters of network integrity in these studies provided a clear representation of the speed and volume of a cerebral network in the recruited subjects. Notably, they proved that some brain regions lost the ability to interact due to the effects of T2DM. Reijmer et al. investigated the network characteristics in T2DM and reported a lower diffusion coefficient in patients [20]. However, Y. Zhang et al. observed higher coefficient in T2DM patients without vascular complications [11]. The other two studies detected no differences in local parameters in T2DM patients compared to healthy controls [36,38].

Major voxels with reduced microstructural integrity were observed mainly in the temporal lobe, frontal lobe and parietal regions of cortex and subcortex in individuals with T2DM. Some studies conducted a whole-brain comparison of DTI metrics by voxel-based analysis and confirmed significant interruption in the integrity of tissue microstructure of the temporal and frontal lobes. Specifically, T2DM patients showed reduced FA in the left, right or bilateral white matter of frontal lobe [30,34,39,40,41]. Additionally, the decreased nodal efficiency of temporal, frontal and parietal lobes in the topological network analysis was reported by two authors [38,42]. Various methods of analysis showed microstructural disruption of the prefrontal subcortical and cortical regions in T2DM patients [8,43,44,45]. Rofey et al. revealed a reduction of FA and the microstructural disruptions in prefrontal white matter and striatum using tractography [27]. Likewise, Fang et al. reported the disruption in frontal-cerebellar pathways and few abnormal microstructural clusters were identified within the parietal lobe [12]. Correspondingly, several studies detected decreased FA in parietal white matter in T2DM patients [8,38,46]. Additionally, higher MD values were observed in parietal cortex in T2DM patients [8,45]. In the temporal lobe, microstructural changes were detected including some structures with lower DTI parameters in T2DM patients [30,38,39,41,46]. These structures include hippocampus, the temporal stem and fusiform gyrus [8,30,43,45]. Moreover, lower hippocampal white matter connections with the regions of frontotemporal lobes were found in T2DM patients compared to healthy participants [47].

In T2DM patients, the studies showed abnormalities in specified subregions of the corpus callosum, including genu, body and splenium [29,32,37,38,48]. Impaired white matter integrity in T2DM patients was found in corona radiata, corticospinal tracts and internal capsules and cingulum, which showed the effects of diabetes [29,32,34,48,49]. Additionally, diabetes was associated with changes in the internal capsule, including the anterior, posterior and the retro lenticular part [29,31,32,38,40] Reduced FA, increased AD, RD or MD in the forceps major, minor and arcuate fasciculus, superior frontal occipital fasciculus and optic radiation were reported in T2DM patients compared to non-diabetic controls [32,35,36,39,45,46].

In the subcortical regions, thalamus, fornix and thalamic radiations were reported to have lower integrity in few studies. One of these studies showed lower significant FA values in the tracts between the dorsal lateral side of the prefrontal cortex in diabetic and obese participants, moreover; the results suggest the presence of an association between T2DM and the changes of hippocampus structure and related limbic structures [28]. Other results showed that decreased FA was found in the hippocampus [38,47] and the corona radiata and cingulum [48]. In the same regions, higher MD in patients with T2DM was observed [43,47,48]. Additionally, increased MD in T2DM patients was observed in the cingulate gyrus and parahippocampal gyrus [41,43]. Network analysis also showed a decrease in nodal efficiency for the hippocampus, amygdala [42] and cingulate gyrus [38], which may suggest brain network abnormalities in T2DM patients.

In the cerebellum, T2DM patients were found to have decreased white matter FA values in the cerebellum, specifically, the peduncles and vermis [34,39,46,50]. Likewise, the cerebellar grey matter was also found to have differences in microstructural integrity [41,45]. Moreover, Fang et al. examined fibre connectivity and found lower connectivity in T2DM patients, and this reduced connectivity predominantly affected the cerebellar circuits [12].

#### 3.2.2. Microstructural and Cognitive Abnormalities in T2DM

The findings of several included studies suggest an association between T2DM and general cognitive deficits [30,38,45,47,51]. Reduced cognitive performance was noticed on different cognitive domains including working memory [8,30,32,38,45,46,47,48,51], attention [8,32,46,51], the speed of information processing [38,52], and executive function [8,32,51]. Surprisingly, some studies mainly demonstrated that the cognitive abnormalities may occur at early diabetic stages or may not be noticed even in old patients and after a long duration of illness [37,38,44]. Xiong et al. demonstrated that T2DM patients with cognitive impairments had more microstructural changes noticed in the cingulum, corona radiata, the frontotemporal region, external and internal capsules, corticospinal tract, and thalamic radiation [16].

Other studies concluded that low FA values in the cingulum were negatively associated with verbal memory deficits [51]. Additionally, increased MD values in the inferior longitudinal fasciculus and parahippocampal gyrus were positively related to verbal memory deficits [37,45,47]. In addition, one study reported that fewer white matter connections were seen in the area between hippocampus and temporal lobe in patients with memory deficits [47]. J. Zhang et al. found that the lower performance of working memory in T2DM patients had a positive correlation with the network of hippocampus and superior temporal lobe [36].

The executive dysfunction in T2DM patients was found to be related to network disorganization and a reduced external capsule integrity compared to healthy controls [32,38]. Zhang et al. demonstrated a significant correlation between the increased MD and the reduced FA of the prefrontal white matter with the attention network test [11]. Additionally, the same study found some significant associations between attention and local nodal efficiency of the temporal lobe [38].

The association between MD and FA values of inferior longitudinal fascicules and corpus callosum and the scores of substitution test suggested an impaired information-processing speed in patients [9]. Moreover, the FA values in grey matter volume and the speed of information processing were negatively correlated in patients [9]. Lastly, Yoon et al. found a relationship between the disorganised network and slower information processing speed (Figure 10a–c) [10].

#### 3.2.3. Microstructural Alterations and Metabolic Profile in T2DM

The findings suggested that the severity of microstructural alterations in brain regions and structures including the frontal lobe, temporal lobe, prefrontal region, praecuneus, cerebral crus, cerebral lobule, superior frontal gyrus were associated with longer duration of diabetes [41,44]. However, a negative correlation between the duration of illness and FA values in white matter was reported [29,52]. Some microstructural changes in specific white matter structures such as both longitudinal fasciculi, cingulum, corpus callosum and uncinate fasciculus did not have any association with the illness duration [7,33,51]. Moreover, network studies confirmed no relationships between the parameters of brain network and the duration of illness [7,36,38,42].

Some studies reported that HbA1c has a negative relationship with the reduction of global microstructural tissue integrity [29] and specifically in inferior longitudinal fasciculus, thalamic radiation, external capsule, internal capsule, fornix, and corpus callosum [36]. In network organization of T2DM patients, HbA1c was positively correlated with shortest path length, and negatively correlated with global efficiency [36]. However, other included articles reported no associations between the microstructural integrity/network organization and HbA1c [33,35,36,38,42,50,52]. In terms of FBS, few included studies reported an association between the brain tissue changes and FBS in the internal and external capsules, cingulum, inferior lobule, splenium, fornix and tracts of the frontal lobe [35,36,47,51]. (Figure 11a,b).

## 4. Discussion

### 4.1. T1DM Patients versus Healthy Controls

The reviewed studies reveal a relationship between the brain tissue changes and cognitive disability in T1DM, and the widespread FA reduction observed in DTI-T1DM studies is in line with a study published by [53]. The reduced diffusivity in the thalamus in adolescent patients is associated with T1DM and glycaemic exposure; also, this observation has an agreement with other studies of adults [54] and youth [55]. The lower scores on the digit span test in adolescent patients are associated with lower FA at the midbrain and optic radiation, which indicate more impaired memory and attention performances. In addition, the visual memory impairment in adolescent patients with T1DM are found to agree with another study and meta-analyses [2,4,55].

Studies conducted on adults with T1DM confirmed that reduced FA values in posterior corona radiata and optic radiation are correlated with hyperglycemic exposure [26]. In contrast, the same result is not confirmed in adolescents with T1DM due to shorter disease duration or lesser exposure to hyperglycemia [22]. Hyperglycemia could increase the risk of brain tissue damage or dysfunction by a metabolism-dependent cascade such as hyperglycemia-induced oxidative stress. The DTI alterations presented in this review could alternatively be due to other glycolysis-independent biological factors that are changed simultaneously with severe hyperglycemic episodes [26].

The observed decreased FA values in various structures of the brain may reflect a complex involvement of tissue properties, and they may suggest white matter disruption, axonal damage, degeneration or irregularity of axonal structure. These low DTI measurements are found to be correlated with reduced neurocognitive performance indicating for the first time that brain microstructural abnormalities may underlie cognitive dysfunction [2]. Kodl et al. confirmed that there is a link between reduced FA in the white matter [24] and few regions with reduced cortical thickness in middle-aged adults [7]. These findings suggest that long-standing T1DM may cause microstructural changes in the posterior cerebrum. In addition, the reduced FA values in the splenium of the corpus callosum, optic radiations, and the posterior corona radiata with posterior cortical atrophy are found to be correlated with volume loss in the occipital cortex in T1DM patients, which may suggest the effects of T1DM on executive functions and psychomotor speed [25].

In children with T1DM, the microstructural differences in white matter are more apparent among those with higher HbA1c. Additionally, the lack of significant differences in FA between diabetics and controls may be due to young age and shorter disease duration of children [20]. However, children with T1DM who have lower AD values are found to have axonal damage and less axonal coherence, which is in agreement with Barnea-Goraly et al. Within the same group, the significant positive correlation between HbA1c and RD indicates that higher blood glucose may affect fibre myelination [6]. Lower FA and higher RD values and the reduced AD and RD values in patients are likely due to the domination of age effect, which significantly correlates with white matter structures and disease duration. Additionally, there is a notable difference between this study and another study of DTI in children with T1DM [20]. The latter found a significant relationship between lifetime HbA1c values and white matter changes, whereas Barnea-Goraly et al. saw only significant associations with HbA1c values at the study baseline [6]. These findings in a large size study may suggest that HbA1c values at baseline are more representative of diabetic complications on the brain than lifetime HbA1c values.

### 4.2. T2DM Patients versus Healthy Controls

The reviewed studies demonstrate that there is a significant association between T2DM and low global integrity and network organisation of brain microstructures. Additionally, they showed that T2DM micro-vascular changes in the brain had been linked to cognitive deficits, supporting the hypothesis that neuroinflammation and microvascular pathology are involved in the cerebral complications/changes of diabetes, which DTI is sensitive to compared with other MRI conventional techniques [56,57]. The findings are in agreement with a previous studies that show the relationship between T2DM and abnormalities in temporal lobe; particularly, in the hippocampus [44]. Similarly, included DTI studies revealed the relationship between microstructural changes and network impairment in temporal lobe and hippocampus, which are responsible for grasping new information and learning process [58]. The results from the reviewed studies demonstrated that abnormalities in the microstructure of the hippocampus and the temporal white and grey matters are seen to be linked to dysfunctionalities in memory.

Another agreement is seen with a previous study in which decreased volume of the prefrontal area is confirmed in patients with T2DM and correlates with higher values of HbA1c [59]. Moreover, the prefrontal white matter and cortex abnormalities are shown to have a significant relationship with diabetes duration. They may underlie attention and memory deficits and executive dysfunction in patients with T2DM. This association signifies that these defects may be caused by the diabetic impact on prefrontal white matter.

Due to the complicated environment of brain networks and white matter tracts, it is difficult to determine the association between white matter disruptions and cognitive functions. However, preliminary results of DTI tractography and network studies suggest that diabetes may affect white matter major tracts. In the studies investigated, the brain network, the nodal efficiency and diffusivity are found to be correlated with various cognitive domains in temporal and prefrontal lobes. Some of the included findings display the association between the impaired microstructure of the thalamus and disorganised brain network in the frontal cortex. Additionally, changes in frontal-striatum thalamic region in patients are found to be correlated with cognitive deficits (executive function, memory, attention and information processing speed) and higher levels of HbA1c [60]. Lastly, DTI-T2DM studies show that the white and grey matter of cerebellum are disrupted in patients, and all patients with longer duration of illness are observed to have more abnormalities in cerebellar connections, which may significantly affect the cognitive performance, mainly executive functions.

### 4.3. Common Microstructural Alterations, Cognitive Deficits and Comorbidities

The DTI studies reviewed here demonstrate that T1DM and T2DM patients have brain microstructural changes in comparison to healthy volunteers. According to the reported findings, there are common regions of microstructural changes in diabetic brains including internal and external capsules, corona radiata, hippocampus, thalamus, inferior and superior longitudinal fasciculi, cerebellum, and corpus callosum (Figure 12). The involvement of microstructural changes in the diabetic brains is linked with observed cognitive dysfunction, including the speed of information processing, memory, executive functions and attention in individuals with T1DM and T2DM. These reported microstructural and cognitive deficits are mainly correlated with uncontrolled glycaemia in both types of diabetes. Regarding the comorbidities, the presence of microangiopathy in T1DM patients, significantly increases the severity of microstructural abnormalities in the brain compared to other T1DM patients. Similarly, T2DM patients with other comorbidities such as obesity, stroke and hypertension increase the seriousness of microstructural changes in the brain.

### 4.4. Multi-Shell Diffusion Imaging Models

A sufficient number of studies suggest that DTI is sensitive in detecting microstructural changes in T1DM and T2DM patients, and it is more sensitive to white matter disruptions than grey matter abnormalities. Nonetheless, DTI metrics could be misleading when imaging complex crossing fibres [39,42]. The lack of specificity in DTI metrics has encouraged the development of advanced diffusion methods, and these methods yield higher sensitivity and specificity to subtle brain changes. Diffusion Kurtosis Imaging (DKI) and Neurite Orientation Dispersion and Density Imaging (NODDI) have been recently used to investigate microstructural changes in T2DM, but not T1DM [48,49].

Unlike DTI, DKI can estimate the Gaussian distribution of water molecules. This technique allows calculation of kurtosis parameters, including mean kurtosis, radial kurtosis, and axial kurtosis, which reflect more structural complexity [61]. In T2DM only, two studies have assessed brain microstructural integrity using DKI and have confirmed that DKI has more sensitivity to brain microstructural changes than DTI, predominantly in the temporal white matter. DKI superiority over DTI may be due to its ability to assume the non-gaussian water diffusion, which is noted to be more robust [62]. In addition to the non-gaussian assumption, DKI metrics are not affected by the restricted crossing fibres in white matter and can accurately detect the grey matter microstructural alterations.

NODDI is used to adopt a diffusion tissue model which can be used to distinguish three types of microstructural environment including CSF, extracellular and intracellular compartments [63]. Unlike DTI and other diffusion models, NODDI has two broad applications, including evaluation of both white and grey matter. In white matter, the orientation dispersion index quantifies complex structures such as the bending and fanning of axons which helps map the brain connectivity. In grey matter, neurite index quantifies sprawling dendrites in the brain, which provides accurate quantifications of grey matter complexity compared to other diffusion models [63]. A recent study showed significant differences in NODDI parameters between T2DM patients and healthy participants, suggesting that NODDI is a specific imaging biomarker for early detection and monitoring of microstructural alterations in the white matter of T2DM patients compared to DTI [49].

### 4.5. Limitations

The reviewed DTI studies for both types of diabetes have some limitations. First of all, the majority of the studies recruited fewer than 80 diabetic patients with different clinical characteristics. Second, the clinical information such as the age of disease onset, cardiovascular disorders, history of metabolic treatment, and blood pressure were not well described in all studies. Third, the small sample size and the differences in patients’ clinical features may affect the result consistency, which may lead to an unsatisfactory conclusion. Fourth, some of the reviewed studies evaluated FA only, but they did not take into account other DTI parameters, including RD, MD and AD. Fifth, several DTI studies of T1DM reported only once timed HbA1c, unlike other studies which additionally reported a lifetime HbA1c of patients. Lastly, this review lacks the quantitative synthesis, which is vital to draw a reliable conclusion about the microstructural alterations in T1DM and T2DM. However, there are several differences in the methods of analysing DTI images which made the implementation of meta-analysis unfeasible.

### 4.6. Future Directions

Simultaneous assessments of white and grey matter alterations would be interesting, and well-designed longitudinal studies are needed to recognise the mechanism of the diffusion changes as the disease progresses. Additional cross-sectional studies will help to confirm whether FA reduction correlates with the duration of diabetes independent of patients’ ages. Furthermore, functional MRI (fMRI) and DTI observational studies are essential to explore the brain structural and functional connectivity in T1DM and to confirm the previous observations in T2DM studies. Finally, additional observational studies are needed to examine and confirm the applicability of NODDI and DKI in quantifying the microstructural abnormalities in T1DM and T2DM.

## 5. Conclusions

DTI provides imaging biomarkers for microstructural alterations that occur in both types of DM. Moreover, the different analysing methods of DTI parameters have assisted researchers in reporting disrupted integrity in the white and grey matters, and network disorganization of cortical and subcortical areas in both types of diabetes. DM microstructurally affects the four major lobes of the brain and major structures such as corpus callosum, thalamus, corona radiata, hippocampus, external and internal capsules. These detected microstructural changes in diabetic brains are associated with cognitive impairments, including attention, executive functions, memory and information-processing speed. There is an association between the clinical profiles of patients with T1DM and T2DM (i.e., disease duration and HbA1c) and altered DTI metrics in various regions of the brain. However, longitudinal DTI studies from early to advanced stages of T1DM and T2DM should be considered to provide a robust conclusion and better understanding of the neural damage in diabetes.

## Figures and Tables

**Figure 1 brainsci-11-00140-f001:**
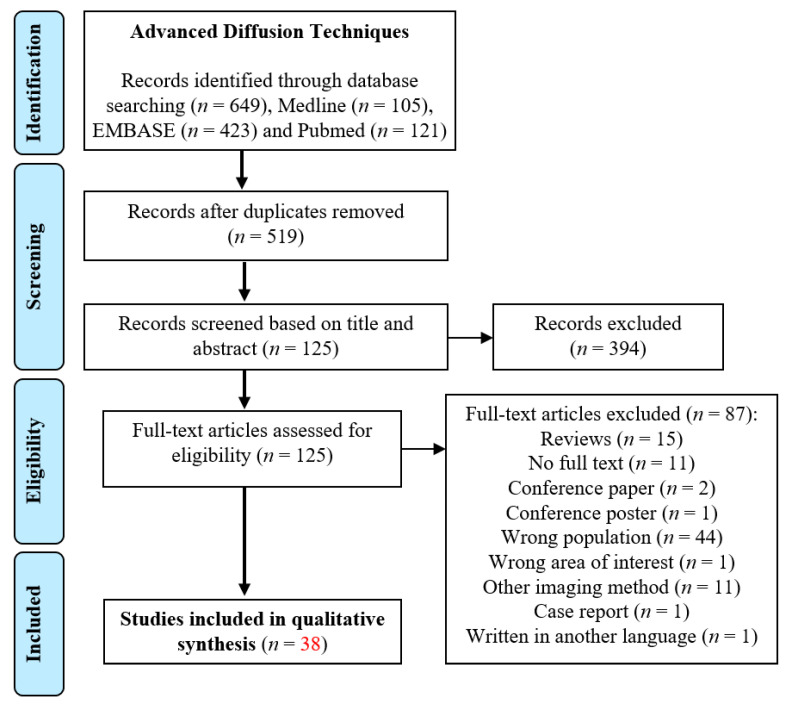
Systematic search strategies including study identifications, screening, eligibility and inclusion.

**Figure 2 brainsci-11-00140-f002:**
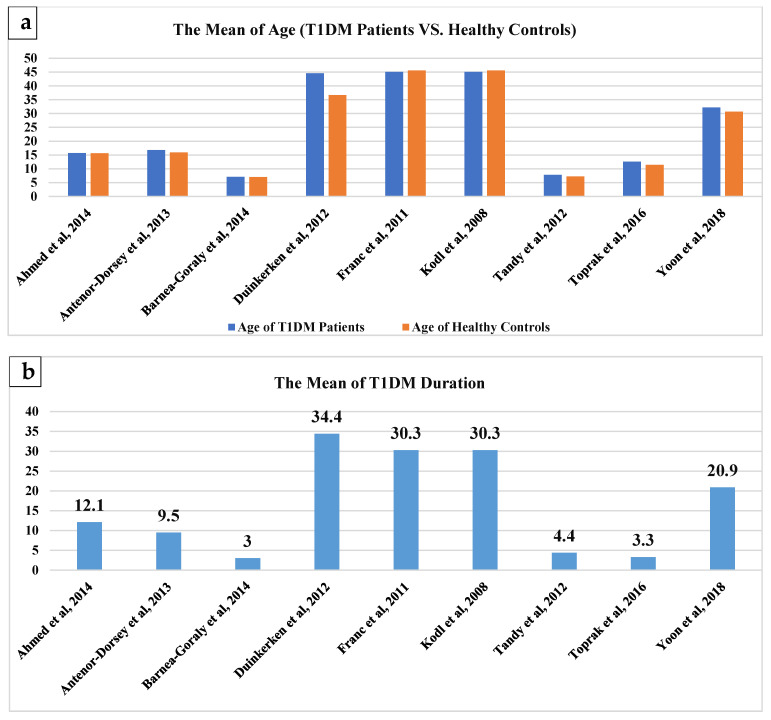
Mean values of clinical characteristics. (**a**) represents the mean age of Table 1 DM and healthy controls; (**b**) represents the mean duration of T1DM; (**c**) represents the mean HbA1C of T1DM and healthy controls; (**d**) represents the mean FBS of T1DM [1,2,3,4,5,6,7,8,9].

**Figure 3 brainsci-11-00140-f003:**
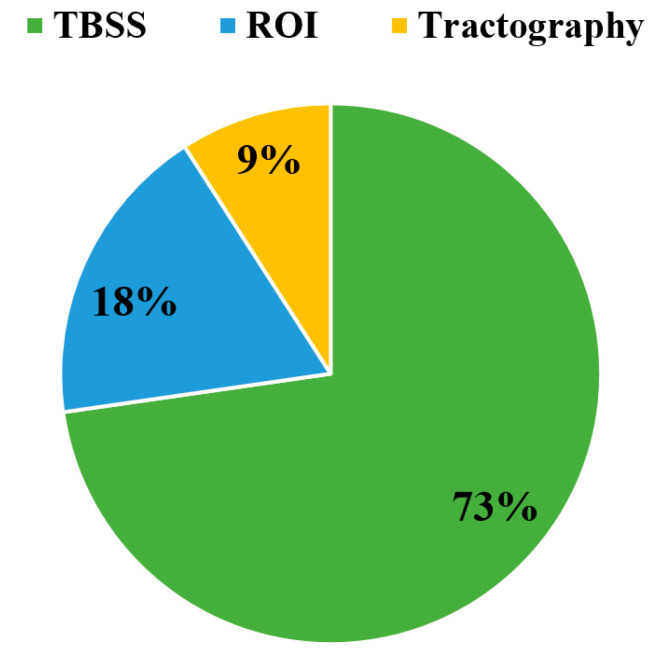
Methods used to analyse brain microstructural changes in T1DM patients.

**Figure 4 brainsci-11-00140-f004:**
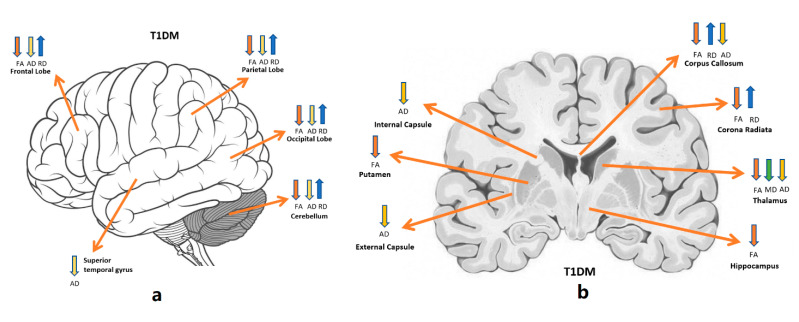
Altered diffusion measures in T1DM brain structures. (**a**) shows the affected brain lobes and cerebellum, (**b**) a cross sectional view for deep brain tissues. Orange: FA; green: MD; yellow: AD; blue: RD. Directions: down: decrease; up: increase [27].

**Figure 5 brainsci-11-00140-f005:**
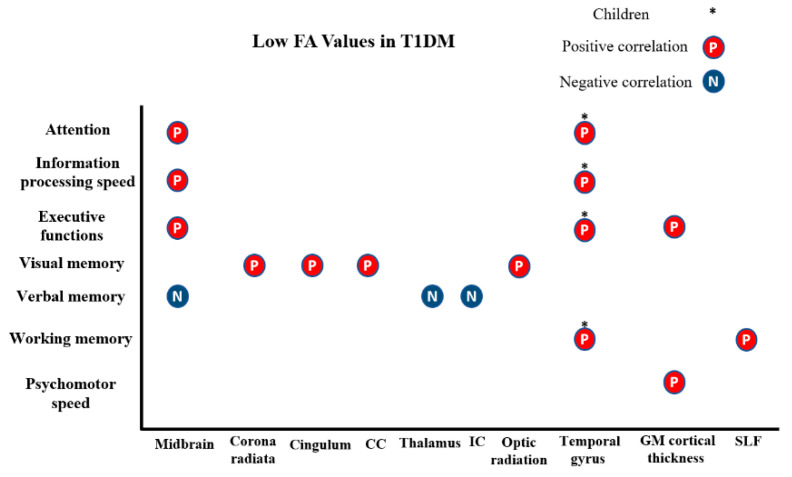
shows the association of FA diffusion metrics with cognitive functions in T1DM. (CC: corpus callosum, IC: internal capsule, and GM: grey matter, SLF: superior longitudinal fasciculus).

**Figure 6 brainsci-11-00140-f006:**
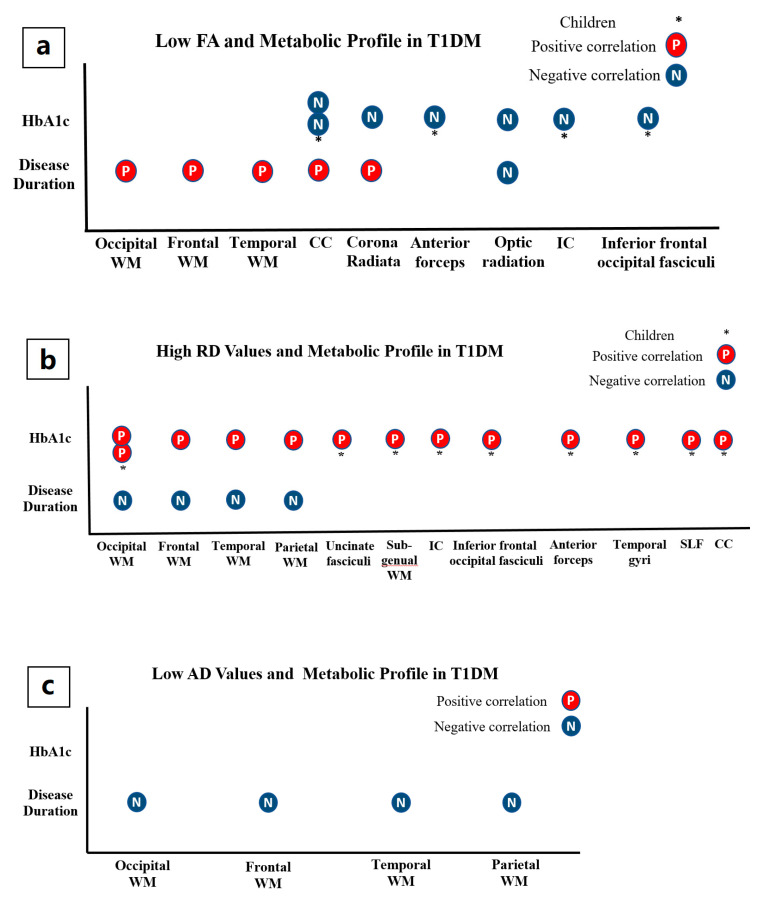
shows the association of diffusion metrics and metabolic profile in T1DM. (**a**) illustrates the correlation between FA and metabolic profile; (**b**) illustrates the correlation between RD and metabolic profile; (**c**) illustrates the correlation between AD and metabolic profile. (CC: corpus callosum, IC: internal capsule, and WM: white matter, SLF: superior longitudinal fasciculus).

**Figure 7 brainsci-11-00140-f007:**
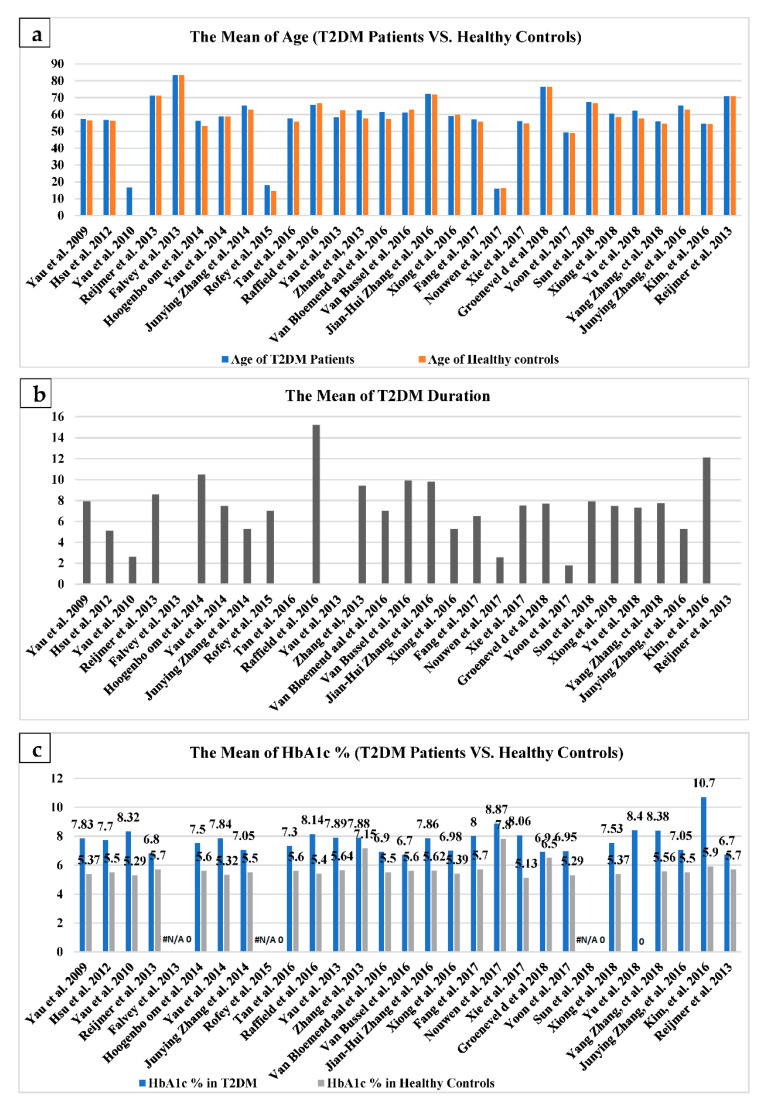
Mean values of clinical characteristics. (**a**) illustrates the mean age of type 2 diabetes mellitus (T2DM) and healthy controls; (**b**) illustrates the mean duration of T2DM; (**c**) illustrates the mean HbA1C of T2DM and healthy controls; (**d**) illustrates the mean fasting blood sugar (FBS) of T2DM [10,11,12,13,14,15,16,17,18,19,20,21,22,23,24,25,26,27,28,29,30,31,32,33,34,35,36,37,38].

**Figure 8 brainsci-11-00140-f008:**
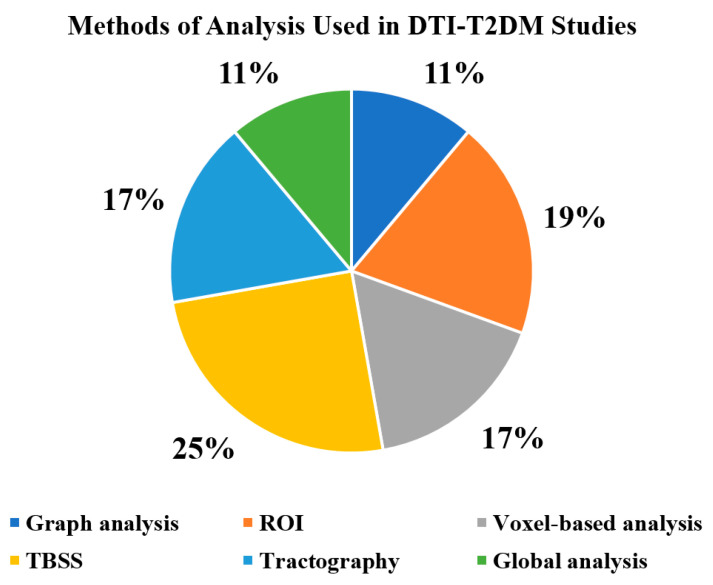
Methods used to analyse brain microstructural changes in T2DM patients. (ROI: region of interest, TBSS: tract based spatial statistics, T2DM: type 2 diabetes mellitus).

**Figure 9 brainsci-11-00140-f009:**
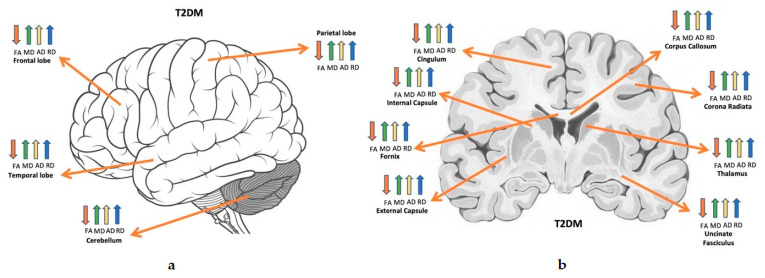
Altered diffusion measures in T2DM brain structures. (**a**) shows the affected brain lobes and cerebellum, (**b**) a cross sectional view for deep brain tissues. Orange: FA; green: MD; yellow: AD; blue: RD. Directions: down: decrease; up: increase [27].

**Figure 10 brainsci-11-00140-f010:**
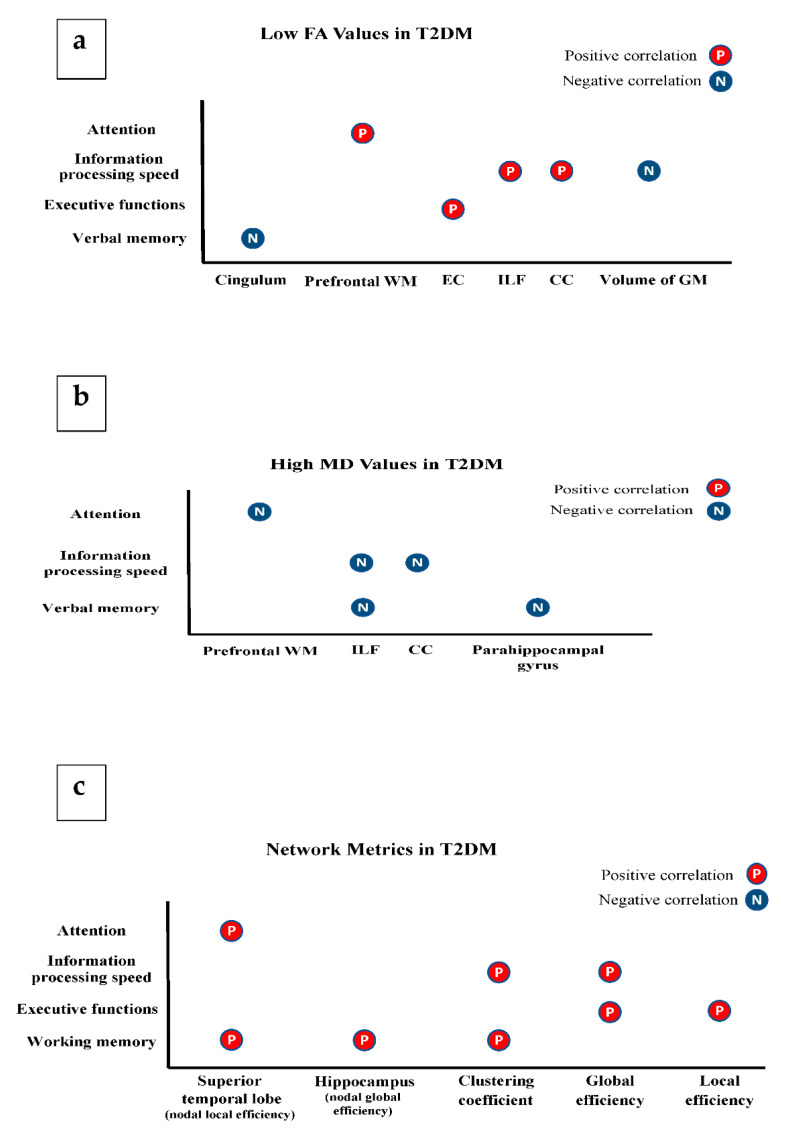
Association of diffusion metrics with cognitive functions in T2DM. (**a**) illustrates the association between FA and cognitive functions; (**b**) illustrates the association between MD and cognitive functions; (**c**) illustrates the association between network parameters and cognitive functions. (CC: corpus callosum, EC: external capsule, and WM: white matter, ILF: inferior longitudinal fasciculus).

**Figure 11 brainsci-11-00140-f011:**
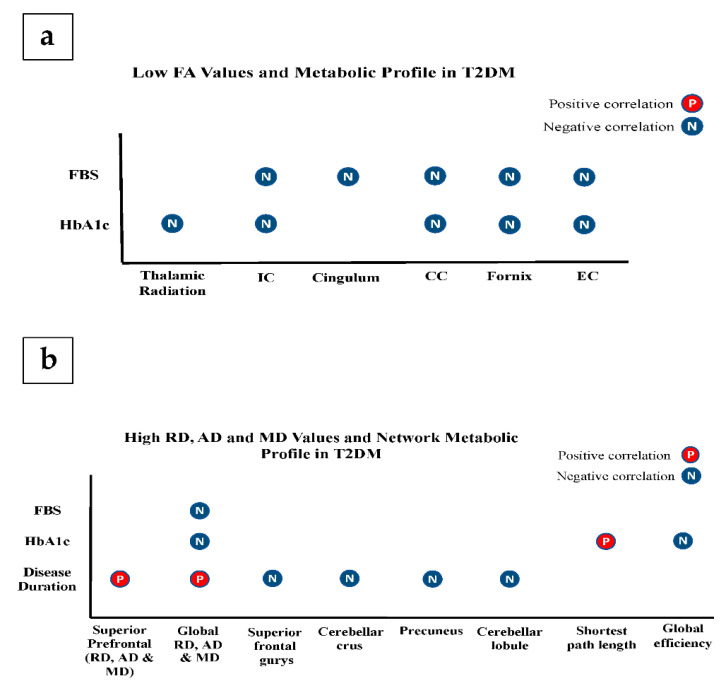
Association of diffusion metrics with metabolic profile. (**a**) represents the association between FA and metabolic profile; (**b**) represents the association between MD, RD, AD and network parameters and the metabolic profile. (CC: corpus callosum, EC: external capsule, and IC: internal capsule).

**Figure 12 brainsci-11-00140-f012:**
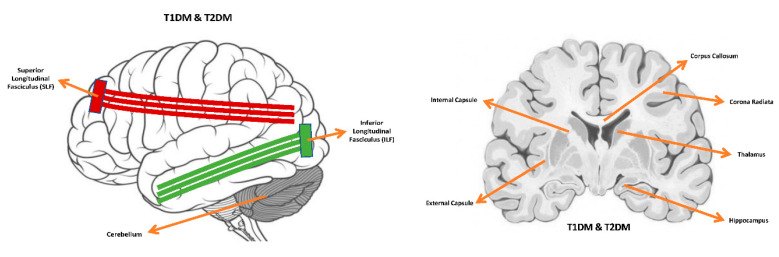
Common microstructural tissue alterations in the brain of patients with type 1 and type 2 [27].

**Table 1 brainsci-11-00140-t001:** Study and participant selection criteria: (**a**) represents the list of study selection and inclusion criteria, (**b**) represents the list of participant selection criteria.

Study Selection and Inclusion Criteria (a)	
Inclusion	Exclusion
Cross-sectional design Case-control design Cohort: Prospective/retrospective	Case reports Reviews or systematic literature reviews Qualitative studies Editorials, comments, news, and letters
Studies published in peer-review journals or the grey literature are considered for inclusion.	
Written in English	
**Participants Selection and Inclusion Criteria (b)**	
**Inclusion**	**Exclusion**
Age: 5–75 years old (flexible)	Neurological disorders and other diseases not due to diabetes Animal or in-vitro
Healthy controls: No neurological disorders No history of diabetes	
Patients: People with type 1 diabetes People with type 2 diabetes	

**Table 2 brainsci-11-00140-t002:** Illustrates the diffusion findings between the groups of T1DM. (↓) increased values and (↑) decreased values.

Microstructural Alterations in Type 1 Diabetes Mellitus (T1DM) Patients				
Region	↓ FA (Fractional Anisotropy)	↓ MD (Mean Diffusivity)	↑ RD (Radial Diffusivity)	↓ AD (Axial Diffusivity)
Corpus callosum	van Duinkerken et al. 2012 (with/without microangiopathy), Kodl et al. 2008, Yoon et al. 2018 [5,7,8].	--	van Duinkerken et al. 2012 (with/without microangiopathy) [8].	Antenor-Dorsey et al. 2013, van Duinkerken et al. 2012 (with/without microangiopathy), Tandy et al. 2012 [1,4,8].
Corona radiata	van Duinkerken et al. 2012 (without microangiopathy), Kodl et al. 2008, Franc et al. 2011, Toprak et al. 2016 [2,7,8].	--	van Duinkerken et al. 2012 (with/without microangiopathy) [8].	--
Thalamic radiation	van Duinkerken et al. 2012 (with microangiopathy) [8].	--	--	van Duinkerken et al. 2012 (with/without microangiopathy) [8].
Forceps minor and major	van Duinkerken et al. 2012 (with microangiopathy) [8].	--	van Duinkerken et al. 2012 (with/without microangiopathy) [8].	van Duinkerken et al. 2012 (with/without microangiopathy) [8].
Superior longitudinal fasciculi (SLF)	van Duinkerken et al. 2012 (with microangiopathy), Ahmed et al. 2014, Yoon et al. 2018 [5,8,9].	--	--	van Duinkerken et al. 2012 (with/without microangiopathy) [8].
Optic radiation	Ahmed et al. 2014, Kodl et al. 2008, Franc et al. 2011 [3,7,9].	--	--	--
Putamen	Toprak et al. 2016 [2]	--	--	--
Inferior longitudinal fasciculus (ILF)	Toprak et al. 2016 [2]	--	--	--
Thalamus	Toprak et al. 2016 [2]	Antenor-Dorsey et al. 2013 [4]	--	Antenor-Dorsey et al. 2013, Tandy et al. 2012 [1,4]
Hippocampus	Toprak et al. 2016 [2]	--	--	--
Inferior frontal-occipital fasciculus	Yoon et al. 2018 [5]	--	van Duinkerken et al. 2012 (with/without microangiopathy) [8]	--
Superior parietal lobule	Antenor-Dorsey et al. 2013 [4]	--	Antenor-Dorsey et al. 2013 [4]	Antenor-Dorsey et al. 2013 [4]
Cerebellum	--	--	Antenor-Dorsey et al. 2013 [4]	Antenor-Dorsey et al. 2013 [4]
Internal capsules	--	--	--	Antenor-Dorsey et al. 2013; Tandy et al. 2012 [1,4]
External capsules	--	--	--	Antenor-Dorsey et al. 2013 [4]
Cerebral peduncle	--	--	--	Antenor-Dorsey et al. 2013 [4]
Cingulate gyrus	--	--	--	Tandy et al. 2012 [1]
Superior temporal gyrus	--	--	--	Tandy et al. 2012 [1]
White matter (widespread changes)	Barnea-Goraly et al. 2014 [6]	--	Barnea-Goraly et al. 2014 (reduced RD with longer disease duration, and increased RD with younger age at disease onset) [6].	Barnea-Goraly et al. 2014 [6]

**Table 3 brainsci-11-00140-t003:** Illustrates the diffusion findings between groups in T2DM.

Microstructural Alterations in T2DM Patients				
Region	↓ FA	↑ MD	↑ RD	↑ AD
Corpus callosum	Yau et al. 2013; Zhang et al. 2014; Tan et al. 2016; Zhang et al. 2016; Nouwen et al. 2017; Yoon et al. 2017; Xie et al. 2017; Sun et al. 2018; Yu et al. 2018, Kim et al. 2016 [10,13,17,18,19,23,26,28,29,36].	Reijmer et al. 2013; Zhang et al. 2014; Xie et al. 2017; Sun et al. 2018; Xiong et al. 2018 [13,18,20,28,38].	Zhang et al. 2014; Nouwen et al. 2017; Xie et al. 2017; Sun et al. 2018; Yu et al. 2018 [13,18,23,28,29].	Reijmer et al. 2013; Xie et al. 2017; Sun et al. 2018 [13,20,28].
Corona radiata	Zhang et al. 2014; Yoon et al. 2017; Xiong et al. 2018; Sun et al. 2018 [10,13,18,38].	Zhang et al. 2014; Xiong et al. 2016; Sun et al. 2018; Xiong et al. 2018 [13,18,23,28,38].	Zhang et al. 2014; Xiong et al. 2016; Sun et al. 2018; Nouwen et al. 2017 [13,16,18,23].	Xiong et al. 2016; Sun et al. 2018 [13,16].
Cingulum	Yau et al. 2010; Hoogenboom et al. 2014; Zhang et al. 2014; Tan et al. 2016; Nouwen et al. 2017; Yoon et al. 2017; Sun et al. 2018; Xion et al. 2018; Zhang et al. 2016; Xiong et al. 2016 [10,13,16,18,23,26,30,31,36,38].	Falvey et al. 2012; Zhang et al. 2014; Sun et al. 2018; Xiong et al. 2018 [13,16,18,34].	Zhang et al. 2014; Xiong et al. 2016; Nouwen et al. 2017; Sun et al. 2018 [13,16,18,23].	Tan et al. 2016; Xiong et al. 2016; Sun et al. 2018 [13,16,26].
Uncinate Fasciculus	Reijmer et al. 2013; Hoogenboom et al. 2014; Zhang et al. 2014; Tan et al. 2016; Nouwen et al. 2017; Yoon et al. 2017; Xiong et al. 2018 [10,18,23,25,26,30,38].	Reijmer et al. 2013; Tan et al. 2016; Xiong et al. 2018 [20,26,38].	Reijmer et al. 2013; Zhang et al. 2014; Nouwen et al. 2017 [16,18,23].	Reijmer et al. 2013 [20]
Superior longitudinal fasciculi (SLF)	Yau et al. 2014; Zhang et al. 2014; Tan et al. 2016; Zhang et al. 2016 [18,22,26,36].	Reijmer et al. 2013; Zhang et al. 2014 [18,20].	Reijmer et al. 2013; Zhang et al. 2014 [18,20].	Reijmer et al. 2013; Bloemendaal et al. 2016 [20,33].
Inferior longitudinal fasciculus (ILF)	Zhang et al. 2014; Kim et al. 2016; Tan et al. 2016; Yoon et al. 2017 [10,17,18,26].	Reijmer et al. 2013; Tan et al. 2016 [20,26].	Reijmer et al. 2013 [20]	Reijmer et al. 2013 [20]
Thalamus	Zhang et al. 2014; Zhang et al. 2016; Tan et al. 2016; Xiong et al. 2016; Fang et al. 2017; Nouwen et al. 2017; Yoon et al. 2017; Sun et al. 2018 [10,12,13,16,18,23,26,36].	Zhang et al. 2014; Tan et al. 2016 [18,26]	Zhang et al. 2014; Xiong et al. 2016; Nouwen et al. 2017 [16,18,23].	Tan et al. 2016; Xiong et al. 2016; Sun et al. 2018 [13,16,26].
Corticospinal tract	Xiong et al. 2016; Sun et al. 2018; Nouwen et al. 2017; Yoon et al. 2017 [10,13,16,23].	--	Nouwen et al. 2017 [23]	Bloemendaal et al. 2016 [33]
Inferior frontal-occipital fasciculus	Zhang et al. 2014; Zhang et al. 2016; Kim et al. 2016; Nouwen et al. 2017; Yoon et al. 2017 [10,17,18,23,36].	--	Nouwen et al. 2017 [23]	Bloemendaal et al. 2016 [33]
Cerebellum	Yau et al. 2013; Yoon et al. 2017; Tan et al. 2017; Fang et al. 2017; Xiong et al. 2016 [10,12,16,19,26].	Yau et al. 2014; Hsu et al. 2012 [14,22].	Xiong et al. 2016; Hsu et al. 2012 [14,16].	Xiong et al. 2016; Hsu et al. 2012 [14,16].
Internal capsules	Yau et al. 2013; Zhang et al. 2013; Zhang et al. 2014; Kim et al. 2016; Xiong et al. 2016; Nouwen et al. 2017; Sun et al. 2018 [13,16,17,18,19,23,32].	Zhang et al. 2014; Xiong et al. 2016; Xiong et al. 2018 [16,18,28,38].	Zhang et al. 2013; Zhang et al. 2014; Xiong et al. 2016; Nouwen et al. 2017 [16,18,23,32].	Xiong et al. 2016; Sun et al. 2018 [13,16]
External capsules	Yau et al. 2013; Zhang et al. 2014; Kim et al. 2016; Xiong et al. 2016; Nouwen et al. 2017; Yoon et al. 2017; Sun et al. 2018; Xion et al. 2018 [10,13,16,18,19,23,38].	Zhang et al. 2014; Xiong et al. 2016; Xie et al. 2017; Xiong et al. 2018 [16,18,28,38].	Xie et al. 2017; Xiong et al. 2016; Nouwen et al. 2017 [16,23,28].	Xiong et al. 2016; Xie et al. 2017; Sun et al. 2018 [13,16,23,28].
Fornix	Zhang et al. 2014; Zhang et al. 2016; Tan et al. 2016; Xiong et al. 2016; Fang et al. 2017; Nouwen et al. 2017; Yoon et al. 2017; Sun et al. 2018 [10,12,13,16,18,23,26,36].	Zhang et al. 2014; Tan et al. 2016 [18,26].	Zhang et al. 2014; Xiong et al. 2016; Nouwen et al. 2017 [16,18,23].	Tan et al. 2016; Xiong et al. 2016; Sun et al. 2018 [13,16,26]
Hippocampus	Van Bussel et al. 2016, Jian-Hui Zhang et al. 2016, Xiong et al. 2016, Xiong et al. 2018, Falvey et al. 2013 [16,34,35,36,38].	Van Bussel et al. 2016, Xiong et al. 2018, Falvey et al. 2013 [34,35,38].	--	--

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
