# Peer review of "Investigating Brain Microstructural Alterations in Type 1 and Type 2 Diabetes Using Diffusion Tensor Imaging: A Systematic Review"

_brainsci, 2021, doi:10.3390/brainsci11020140_

Round 1

Reviewer 1 Report

This is an interesting review providing important insights on the potential efficiency of DTI for quantification and assessment of brain microvascular abnormalities in diabetic patients.

The authors also carefully discussed the relationships of such changes with cognitive functions and metabolic profile in both type 1 and type 2 diabetes. The manuscript is quite complete. As the Authors stated in the limitations section, despite the lack of quantitative analyses as well as the relatively small sample size in the research studies recruited, this article can put the basis for more specific interventions aimed at improving localization and quantification of neural damage in diabetes.

There are only minor comments that need to be addressed.

Minor comments

1) Page 2, line 46: please justify the text.

2) Figures are not sharp and do not make easy reading. Also, some figures (e.g. figures 2, 5, 6, 7, and 10) need a larger font size.

3) Page 12, line 352-352: please adjust the size of text.

4) Page 17, line 498: please correct the mistake in the sentence.

Reviewer 2 Report

In an elaborate and well-researched literature review, the authors gathered data on white matter tractography and diabetes, making a distinction between Tp1 and Tp 2.

The bibliographic search is well-described.

The principles of DTI are briefly described. The data are interpreted with caution but the authors seem to convince us that some DTI parameters are (not specific for diabetes, I presume, please discuss) related to cognition.

The illustrations are many but add to the completeness of the review.

For the diabetologist and any clinician, the authors might relate the DTI findings to known or suspected pathologies in the brain  resulting from diabetes or its complications such as: Neuroinflammation, glucose transport at the BBB (cerebral insulin resistance), cerebral glucose metabolism, perfusion abnormalities. If such data are not available in the many articles, a theoretical discussion putting the DTI into a more familiar context would be welcome.

Line 25: Diabetic brain is neither an accepted nor an acceptable term.
